# DIFFUSING GRAPH ATTENTION

## ABSTRACT

The dominant paradigm for machine learning on graphs uses Message Passing Graph Neural Networks (MP-GNNs), in which node representations are updated by aggregating information in their local neighborhood. Recently, there have been increasingly more attempts to adapt the Transformer architecture to graphs in an effort to solve some known limitations of MP-GNN. A challenging aspect of designing Graph Transformers is integrating the arbitrary graph structure into the architecture. We propose *Graph Diffuser* (GD) to address this challenge. GD learns to extract structural and positional relationships between distant nodes in the graph, which it then uses to direct the Transformer's attention and node representation. We demonstrate that existing GNNs and Graph Transformers struggle to capture long-range interactions and how Graph Diffuser does so while admitting intuitive visualizations. Experiments on eight benchmarks show Graph Diffuser to be a highly competitive model, outperforming the state-of-the-art in a diverse set of domains.

## 1 INTRODUCTION

Graph Neural Networks have seen increasing popularity as a versatile tool for graph representation learning, with applications in a wide variety of domains such as protein design (e.g., Ingraham et al. (2019)) and drug development (e.g., Gaudelet et al. (2020)). The majority of Graph Neural Networks (GNNs) operate by stacking multiple local message passing layers Gilmer et al. (2017), in which nodes update their representation by aggregating information from their immediate neighbors Li et al. (2016); Kipf & Welling (2017); Hamilton et al. (2017); Veličković et al. (2018); Wu et al. (2019a); Xu et al. (2019b).

In recent years, several limitations of GNNs have been observed by the community. These include under-reaching Barceló et al. (2020), over-smoothing Wu et al. (2020) and over-squashing Alon & Yahav (2021); Topping et al. (2022). Over-smoothing manifests as node representations of well-connected nodes become indistinguishable after sufficiently many layers, and over-squashing occurs when distant nodes do not communicate effectively due to the exponentially growing amount of messages that must get compressed into a fixed-sized vector. Even prior to the formalization of these limitations, it was clear that going beyond local aggregation is essential for certain problems Atwood & Towsley (2016); Klicpera et al. (2019).

Since their first appearance for natural language processing, Transformers have been applied to domains such as computer vision(Han et al. (2022)), robotic control Kurin et al. (2020), and biological sequence modeling Rives et al. (2021) with great success. They improve previous models' expressivity and efficiency by replacing local inductive biases with the global communication of the attention mechanism. Following this trend, the Transformer has been studied extensively in recent years as a way to combat the issues mentioned above with GNNs. Graph Transformers (GTs) usually integrate the input into the architecture by encoding node structural and positional information as features or by modulating the attention between nodes based on their relationships within the graph. However, given the arbitrary structure of graphs, incorporating the input into the Transformer remains a challenging aspect in designing GTs, and so far, there has been no universal solution.

We propose a simple architecture for incorporating structural data into the Transformer, Graph Diffuser (GD). The intuition that guides us is that while the *aggregation* scheme of GNNs is limited, the

*propagation* of information along the graph structure provides a valuable inductive bias for learning on graphs.

Figure 1 shows an overview of our approach. We start with the graph structure (as shown on the left) and construct *Virtual Edges* (middle) that capture the propagation of information between nodes at multiple propagation steps. This allows our approach to zoom out of the local message passing and relates distant nodes that do not have a direct connection in the original graph. We then use the virtual edges to direct the transformer attention (shown on the right) and node representations.

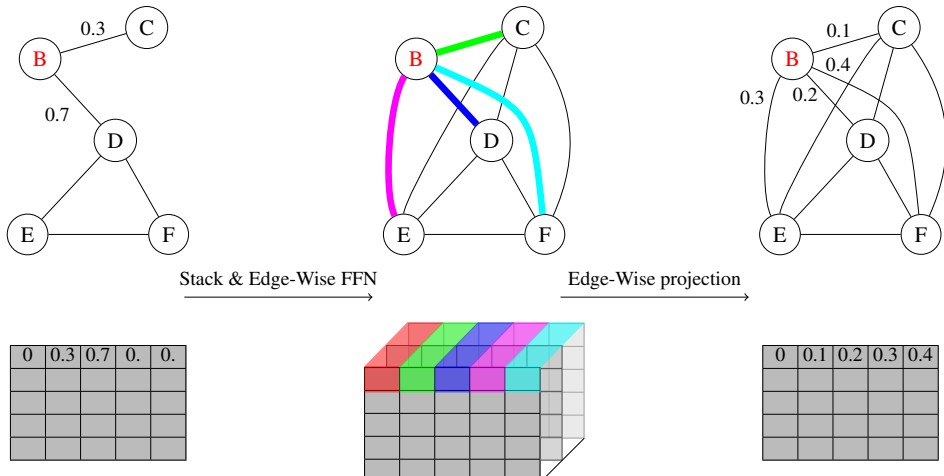

Figure 1: Illustration of our positional attention, focusing on node B. Information propagation from multiple propagation steps is combined to create Virtual Edges(colored) between distant nodes, which then direct the Transformer's attention in each layer.

In the following sections, we first show that existing GNNs and Graph Transformers struggle to model long-range interactions using a seemingly trivial problem. We follow by defining Graph Diffuser and then show how it solves the same problem. Finally, we demonstrate the effectiveness of our approach on eight benchmarks spanning multiple domains by showing it outperforms state-of-the-art with no hyperparameter tuning.

## 2  RELATED WORK

**Message Passing Graph Neural Networks (MP-GNNs)** MP-GNNs Gori et al. (2005); Scarselli et al. (2008) have been the predominant method for graph representation learning in recent years, and have been applied to a wide variety of domains (e.g., Kosaraju et al. (2019); Nathani et al. (2019); Wang et al. (2019); Huang & Carley (2019); Yang et al. (2020); Ma et al. (2020); Wu et al. (2020); Zhang et al. (2020)), MP-GNNss update node representation by stacking multiple layers in which each node aggregates information from its local neighborhoodLi et al. (2016); Kipf & Welling (2017); Veličković et al. (2018); Wu et al. (2019a); Xu et al. (2019b); Hamilton et al. (2017); Xu et al. (2019a). However, as mentioned above, they suffer from under-reaching Barceló et al. (2020), over-smoothing Wu et al. (2020) and over-squashing Alon & Yahav (2021); Topping et al. (2022). Several works have addressed the problem of over-squashing. Gilmer et al. (2017) add "virtual edges" to shorten long distances, and Scarselli et al. (2008) add "supersource nodes". None of these works, however, integrate such virtual nodes or edges into the Transformer. Another line of work uses the attention mechanism Veličković et al. (2018); Brody et al. (2022) to dynamically propagate information in the graph rather than use the original adjacency matrix or Laplacian.

**Graph Transformers (GTs)** Considering their successes in natural language understanding Vaswani et al. (2017); Kalyan et al. (2021),computer vision d'Ascoli et al. (2021); Han et al. (2022); Guo et al. (2021), robotic control Kurin et al. (2020) and a variety of other domains, there have been numerous attempts to apply Transformers to graphs. These works add positional encoding as node features, similar to the encoding in Vaswani et al. (2017), or use relative positioning to bias the attention between nodes, similar to Shaw et al. (2018). Others combine Transformer with

local MP-GNN models by interleaving Rampášek et al. (2022) or stacking them Jain et al. (2021), much in the same way the Transformers were stacked on top of CNNs in computer vision Carion et al. (2020). Early Graph Transformer works Dwivedi & Bresson (2020) used the graph Laplacian eigenvectors as the node positional encoding (PE) to provide a sense of nodes' location in the input graph. SAN Kreuzer et al. (2021) significantly improved this idea by using an invariant aggregation for the eigenvectors. Since then, there have been many recent works constructing PE from the Laplacian Lim et al. (2022); Beaini et al. (2021); Wang et al. (2022); Lim et al. (2022), as well as other structural or positional information such as the node degree Ying et al. (2021) and random-walk SE Dwivedi et al. (2022a). Relative positioning was applied with great success on a large molecular benchmark by Graphormer Ying et al. (2021); Shi et al. (2022) by using pair-wise graph distances to direct the Transformer's attention. Further, GraphiT Mialon et al. (2021) used relative positions between nodes derived from the random walk kernel to bias the attention but heuristically chose a single random walk length. In contrast, Graph Diffuser learns to combine multiple distances. Since then, various works(e.g SAT Chen et al. (2022), EGT Hussain et al. (2021), GRPE Park et al. (2022)) applied the graph structure as a soft bias to the attention. However, all of the works above use the original adjacency matrix or Laplacian to learn positional or relative encoding, unlike this work that learns the adjacency matrix using node and edge features.

**Diffusion** Diffusion and other distance measures were used before to increase the expressivity of MP-GNNs Li et al. (2020b), replace the local message passing scheme( Atwood & Towsley (2016); Wu et al. (2019b); Klicpera et al. (2019; 2018)) or modulate the Transformers attention Mialon et al. (2021). None of these works, however, combines information from multiple different propagation steps.

To the best of our knowledge, this work is the first Graph Transformer to: (1) learn to construct a new adjacency matrix using node and edge features to generate positional or relative encoding, and (2) learn to combine information propagation over multiple different propagation steps in an end-to-end manner.



(a) An example of a $5 \times 5$ grid. Nodes are colored in one out of 20 colors. The goal is to predict, for each node, how many other nodes in the same row or column have the same color. For example, node 23, at the bottom row, has only a single node (node 20) that has the same color(green) and is in the same row or column with node 23. Therefore, its label is 1. The label for node 6 is 2, since it matches with nodes 5 and 21.

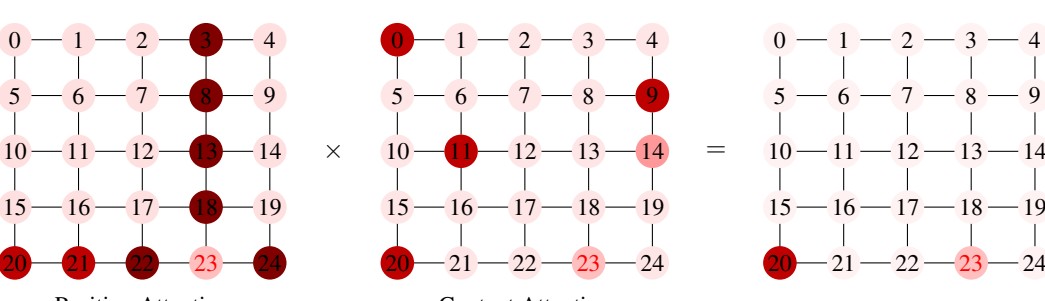

(b) Visualization of the position and content attention our trained model gives to node 23 in the input graph shown in 2a. The model learns to pay attention to nodes in the same row and column using positional attention and to nodes of the same color using content attention. Their element-wise product selects only the relevant nodes for solving the task.

Figure 2: An example of $2D$ Grid Histogram Counting with a $5 \times 5$ grid. 2a shows the original graph. 2b shows the attention patterns our model learns for node 23 in the graph.

# 3 WHY DO WE NEED ANOTHER GRAPH TRANSFORMER?

Given the successful application of Transformers to other domains and the flurry of recent Graph Transformers, it is natural to ask why is there a need for another Graph Transformer?

To illustrate the difficulties of current GNNs and Graph Transformers in modeling interactions in a graph, we use a simple synthetic node classification task. Counting the frequencies of tokens in a sequence is an elementary task that Weiss et al. (2021) showed the original Transformer could easily solve. We propose a simple extension of this task to graphs: In **Grid Histogram Counting** (Figure 2), we generate $N \times M$ grids with randomly colored nodes and ask models to predict, for each node, how many other nodes in the same row or column have the same color. This is a contrived problem but illustrates many real-world problems' needs. Solving it requires far away nodes to communicate, and the communication should consider both the nodes' content (color) and relation within the graph (being in the same row or column).

This is a straightforward generalization of the $1D$ sequential problem that Transformer easily solves to a $2D$ graph. However, as we show in Section 5.1.2, it defeats all existing GNN and Graph Transformer techniques, while Graph Diffuser succeeds.

# 4 GRAPH DIFFUSER

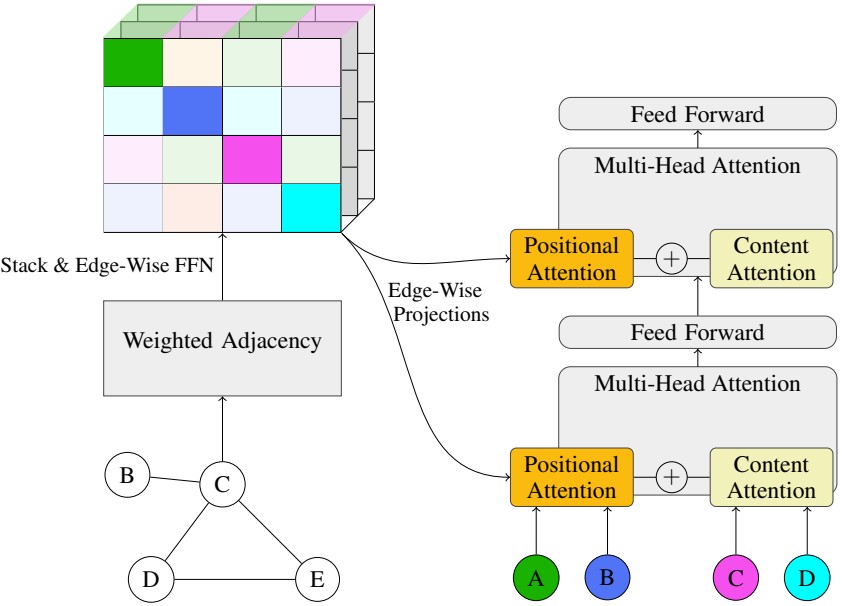

Figure 3: An illustration of Graph Diffuser with 2 Transformer layers. Virtual Edges are created by combining multiple powers of the weighted adjacency matrix(left). Virtual Edges modulate each of the Transformers attention layers(right). Self-Virtual Edges are added to the Transformer input as positional encoding(bottom right)

We now describe our approach for taking a graph $G = (X, A)$ with a node embeddings matrix $X$ and edges $A$ and processing it using the Transformer, which usually takes only a single matrix $X$ as input. Our approach consists of 2 stages, illustrated in the left and right halves of Figure 3. First, GD embeds the structural relations between distant nodes in what we refer to as Virtual Edges. Then, the Transformer processes the nodes while using the virtual edges to direct the computation at different layers.

## 4.1 VIRTUAL EDGES

*Virtual Edges* are high-dimensional representations constructed between distant nodes in the graph. They contain rich information on the structural and positional relationships.

### 4.1.1 POWERS OF THE ADJACENCY MATRIX

To consider relations between distant nodes, the first step is broadening the receptive field on which the architecture operates. In a row-normalized adjacency matrix $A$, $A_{ij}^k$ corresponds to the probability of getting from node $i$ to node $j$ in a $k$ step random walk. We stack different powers of $A$ into a 3 dimensional tensor $E \in R^{n \times n \times k}$

$$E = [I|A|A^2|..|A^k] \tag{1}$$

where $k$ is the number of stacks and $|$ denotes stacking matrices along the 3ed dimension and $I$ is the identity matrix.

Multiplication by the adjacency matrix is closely related to the aggregation scheme of MP-GNNS Xu et al. (2018), however, considering multiple powers of the matrix at once, we zoom out of the local Message Passing paradigm and discover information that GNNs may not detect. Virtual Edges contain structural information such as if nodes are on an odd length cycle, can distinguish many non-isomorphic graphs, and cover many distance measures such as shortest distance and generalized PageRank Li et al. (2020b).

### 4.1.2 EDGE-WISE FEED-FORWARD NETWORK

In order to mix information between different propagation steps and extract meaningful structural relations, each stack is processed by a fully connected edge-wise feed-forward network. Each layer consists of 1 hidden layer with batch norm, ReLU activation, and a residual connection.

$$Edge\text{-}FFN(E_{ij}) = ReLU(BN(E_{ij}W1))W2^1 \tag{2}$$

$$E_{ij} = Edge\text{-}FFN(E_{ij}) + E_{ij} \tag{3}$$

The Edge-Wise FFN consists of 2 such layers and we apply batch norm on the input, before the first layer. This network is applied once, before any of the Transformers layers.

### 4.1.3 WEIGHTED ADJACENCY

Rather than using the original adjacency matrix in equation 1, we found it beneficial to learn a new adjacency matrix using the node and edge features.

$$\hat{A_{ij}} = ReLU(BN([x_i; x_j, e_{ij}]W_1))W_2 \tag{4}$$

$$A_{ij} = normalize(\sigma(\hat{A_{ij}})) \tag{5}$$

Where ; means contacting vectors, $\sigma$ is the sigmoid function and noramlize means $l1$ row normalization. $x_i, x_j \in R^d$, $e_{ij} \in R^{d_{edge}}$, $W_1 \in R^{(2d+d_{edge}) \times 2d}$ and $W_2 \in R^{2d \times 1}$. If there are no edge features, we use only the nodes. This aligns with existing approaches such as GAT (Veličković et al. (2018); Brody et al. (2022)), which find it beneficial to dynamically learn a new adjacency matrix.

## 4.2 INTEGRATING WITH THE TRANSFORMER

Most Graph Transformers use the graph structure to either alter the node's representations or to affect the attention mechanism itself. Graph Diffuser combines the 2 approaches. In the input layer, Self-Virtual Edges are added to the node representation as positional encoding, and at each attention layer, the Virtual Edges are reduced to an attention matrix that is combined with the standard dot-product attention.

### 4.2.1 ATTENTION

At each of the Transformer attention layers, we linearly project each virtual edge $E_{ij}$ separately to get the positional attention score between $i$ and $j$.

$$\hat{Att}_{ij}^{Position} = E_{ij}W_p \tag{6}$$

---

[1]Omitting bias for clearity

We then combine the positional attention and the content(dot-product) attention scores and apply row-wise normalization.

$$\hat{Att}_{ij}^{Content} = \frac{QK^T}{\sqrt{d}}_{ij} \tag{7}$$

$$Att = normalize(exp(\hat{Att}^{Content}) \odot \sigma(\hat{Att}^{Position})) \tag{8}$$

Where $\sigma$ is the sigmoid function and $\odot$ means element-wise multiplication. In multi-head attention with $h$ heads, the projection in equation 6 is done $h$ times.

equation 8 can be viewed as scaling the content attention coefficients based on the positional attention. As we will see in the next section, separating the attention based on content and positions and then combining them seems to assist with learning meaningful connections in the data and provides a natural visualization mechanism.

### 4.3 POSITIONAL ENCODING

Self-Virtual-Edges, $E_{ii}$, are Virtual Edges between a node to itself. They contain important structural information, such as whether the node is part of an odd length cycle and the degrees of adjacent nodes. We project Self-Virtual-Edges to the node's representation dimensionality and add them as positional encoding.

$$X_i = X_i + Relu(E_{ii}W_{pe}) \tag{9}$$

Where $W_{pe} \in R^{k \times d}$.

Self-walks were shown (Dwivedi et al. (2022a)) to be an effective positional encoding. Adding such features to the node representations allows the Transformer to use structural information in all of its layers, including the feed-forward network.

## 5 EXPERIMENTS

### 5.1 GRID HISTOGRAM COUNTING

#### 5.1.1 SETUP

We generate $10k$ grids of size $10 \times k$, where $k \in \{10, 11, 12, 13\}$ with node colors uniformly from a set of 20 colors, and split them into $8k/1k/1k$ train/valid/test sets. The task is cast as a node multi-class classification task, where each node representation is used to predict the number of other nodes in the same row or column with the same color, as illustrated in Figure 2a. More details are in Appendix A.1

#### 5.1.2 RESULTS

Table 1: Grid Histogram Counting average accuracy and s.d over 10 different seeds. *GNN→ Transformer* refers to Transformer blocks stacked on top of GNN (Jain et al. (2021)) and *GNN | Transformer* refers to combining Transformer and GNN blocks in the same layer (Rampášek et al. (2022)).

| Model | Accuracy ↑ | Parameters |
|---|---|---|
| *GNN* | 0.4832 ±0.003 | 68k |
| *Transformer* | 0.44 ±0.003 | 53k |
| *Transformer+RWPE* | 0.4797 ±0.0044 | 53k |
| *Transformer+SignNet* | 0.488 ±0.003 | 72k |
| *GNN → Transformer* | 0.4795 ±0.003 | 44k |
| *GNN | Transformer+RWPE* | 0.534 ±0.0043 | 86k |
| *Graph Diffuser* position only | 0.6073 ±0.011 | 34k |
| *Graph Diffuser* | **0.9755 ± 0.025** | 43k |

Table 1 shows the results. First, we observe that the vanilla Transformer, which easily solves the histogram task over sequential input, fails here and performs the worst. This is not surprising as the Transformer is oblivious to the graph structure. Next, we observe that RWPE Dwivedi et al. (2022a), a popular positional embedding technique, improves very slightly over the base Transformer. This may be because RWPE is unable to detect symmetries in the graph. Nodes 0,4,20 and 24 in the example illustrated in Figure 2a will all get the same embedding by RWPE. Adding SignNet Lim et al. (2022), a theoretically expressive positional embedding technique based on the graph eigenvectors, does not yield much improvement either. This may indicate difficulties learning useful embeddings in eigenvectors based techniques. The GNN baseline, which in theory can solve the task, achieves $0.483$ accuracy. This may be due to the over-squashing phenomenon. Indeed in a $10 \times 10$ grid, passing information between opposing nodes in the same row will require 9 message passing layers to deliver a single message that has been "squashed" with nearly 90k other messages. Next, we observe that using only positional attention(without dot-product attention) performs the second best while having the least parameters. Finally, Graph Diffuser nearly solves the task entirely by combining both positional and content attention. Looking at Figure 2b, we can see the model learns to use the different attention mechanisms in an intuitive way, with content attention detecting nodes in the same color and position attention detecting nodes in the same row or column.

## 5.2 BENCHMARKING GRAPH DIFFUSER

Throughout our experiments, we take an existing Graph Transformer architecture and use it "as is" as our Transformer module without doing any hyper-parameter search. That is, we take a graph Transformer with the same hyperparameters used by the original work and only add our positional attention and encoding.[2] For 5 of the datasets, ZINC, OGBG-{molpcba, ppa,code2}, PCQM-Contact and PCQM4Mv2, we use the very competitive GPS Rampášek et al. (2022) as the Transformer model. For the LRGB benchmarks, Peptides-func and Peptides-struct, there are no official GPS hyperparameters, and we use a Transformer with the authors' baseline hyperparameters and a CLS token added to the input. For a detailed hyperparameters report, see Table 8 and 9.

### 5.2.1 RESULTS

We evaluate our model on 8 datasets overall and compare our results with those of popular MP-GNNs, Graph Transformers and other recent state-of-the-art models. Graph Diffuser exceeds SOTA in 6 of them and achieves very competitive results in the rest. All results for the comparison methods are taken from the original paper or their official leaderboards.

**ZINC** Dwivedi et al. (2020a) is a molecular regression dataset, where the value to be predicted is the molecule's constrained solubility. Table 2 shows the results, where we exceed SOTA results by a substitutional margin.

**Open Graph Benchmark** Hu et al. (2020) is a highly competitive benchmark with a variety of datasets. We evaluate GD on three graph-level prediction tasks from different domains and report the results at Table 3 **OGBG-MOLPCBA** is a multi-task binary classification dataset containing 438k molecules, and the task is to predict the activity/inactivity of 128 properties. Here we rank the second highest and the first among all GNN or Transformer architectures. **OGBG-PPA** consists of protein-protein interaction networks of different species, where the task is predicting the category of species the network is from. The previously highest-ranking model is GPS, and adding our positional attention and encoding to it proves to be an efficient strategy that reaches a new state-of-the-art. **OGBG-CODE2** contains Abstract Syntax Trees(ASTs) of Python methods, and models are tasked with predicting the methods' names. The average distance between nodes in this dataset is larger than that of any other dataset in our experiments, which can explain why Graph Transformers variations dominate its leaderboards. Since positional encoding was not found to be significant by the previous three highest ranking models in this dataset, and considering the large improvement of our model over GPS, it seems that positional attention brings significant benefits in this benchmark, which allows us to reach a new state-of-the-art. **OGB-LSC PCQM4Mv2** Hu et al. (2021) is a large-scale molecular dataset with over 3.7M graphs. Here, our results are similar to GPS, which ranks highest among Graph Transformers variations. Our results lag behind only GEM-2 Liu et al. (2022), which was designed specifically for modeling molecular interactions.

---

[2]We remove any positional encoding used by the original work.

Table 2: MAE for the ZINC dataset from Dwivedi et al. (2020b).

| Model | MAE ↓ |
|---|---|
| GCN Kipf & Welling (2017) | 0.367 ± 0.011 |
| GIN Xu et al. (2019a) | 0.526 ± 0.051 |
| GAT Veličković et al. (2018) | 0.384 ± 0.007 |
| GatedGCN Bresson & Laurent (2017); Dwivedi et al. (2020b) | 0.282 ± 0.015 |
| GatedGCN-LSPE Dwivedi et al. (2022a) | 0.090 ± 0.001 |
| CRaWl Toenshoff et al. (2021) | 0.085 ± 0.004 |
| GIN-AK+ Zhao et al. (2022) | 0.080 ± 0.001 |
| SAN Kreuzer et al. (2021) | 0.139 ± 0.006 |
| Graphormer Ying et al. (2021) | 0.122 ± 0.006 |
| K-Subgraph SAT Chen et al. (2022) | 0.094 ± 0.008 |
| GPS Rampášek et al. (2022) | 0.070 ± 0.004 |
| Graph Diffuser(ours) | **0.0683 ± 0.002** |

**Long Range Graph Benchmark** Dwivedi et al. (2022b) is a recently proposed dataset specifically designed to evaluate models on their ability to capture long-range interactions. We are the first, other than the author's baselines, to evaluate our model on these datasets and currently rank first in all of them. **Peptides-func** and **Peptides-struct** are multi-label graph classification and regression datasets containing 15.5k Peptides molecular graphs. They are of particular interest since the molecules in them consist of many more nodes, on average, than the other molecular datasets in our experiments. As we can see in Table 5, adding our positional attention and encoding outperforms both RWSE and LapPE encodings. **PCQM-Contact** is a molecular dataset with an edge prediction task of predicting if two atoms interact. Our results are reported in Table 6.

Table 3: Test results in OGB graph-level benchmarks Hu et al. (2020). Pre-trained or ensemble models are not included. **Bold**:first, Underlined:Second.

| Model | ogbg-molpcba | ogbg-ppa | ogbg-code2 |
|---|---|---|---|
| | Avg. Precision ↑ | Accuracy ↑ | F1 score ↑ |
| GCN+virtual node Kipf & Welling (2017) | 0.2424 ± 0.0034 | 0.6857 ± 0.0061 | 0.1595 ± 0.0018 |
| GIN+virtual node Xu et al. (2019a) | 0.2703 ± 0.0023 | 0.7037 ± 0.0107 | 0.1581 ± 0.0026 |
| PNA Corso et al. (2020) | 0.2838 ± 0.0035 | – | 0.1570 ± 0.0032 |
| GIN-AK+ Zhao et al. (2022) | 0.2930 ± 0.0044 | – | – |
| DeeperGCN Li et al. (2020a) | 0.2781 ± 0.0038 | 0.7712 ± 0.0071 | – |
| DGN Beaini et al. (2021) | 0.2885 ± 0.0030 | – | – |
| CRaWl Toenshoff et al. (2021) | **0.2986 ± 0.0025** | – | – |
| ExpC Yang et al. (2022) | 0.2342 ± 0.0029 | 0.7976 ± 0.0072 | – |
| SAN Kreuzer et al. (2021) | 0.2765 ± 0.0042 | – | – |
| GraphTrans (GCN-Virtual) Jain et al. (2021) | 0.2761 ± 0.0029 | – | 0.1830 ± 0.0024 |
| K-Subtree SAT Chen et al. (2022) | – | 0.7522 ± 0.0056 | 0.1937 ± 0.0028 |
| GPS Rampášek et al. (2022) | 0.2907 ± 0.0028 | 0.8015 ± 0.0033 | 0.1894 ± 0.0024 |
| Graph Diffuser | 0.2931 ± 0.0034 | **0.8133 ± 0.0057** | **0.1941 ± 0.0014** |

Table 4: Evaluation on PCQM4Mv2 Hu et al. (2021). Since the test set labels are private, we use 150k examples from the train set as our validation and the validation set is treated as the test set.

| Model | MAE ↓ | # Param. |
|---|---|---|
| GCN Kipf & Welling (2017) | 0.1379 | 2.0M |
| GCN-virtual | 0.1153 | 4.9M |
| GIN Xu et al. (2019a) | 0.1195 | 3.8M |
| GIN-virtual | 0.1083 | 6.7M |
| GRPE Park et al. (2022) | 0.0890 | 46.2M |
| EGT Hussain et al. (2021) | 0.0869 | 89.3M |
| Graphormer Shi et al. (2022) | 0.0864 | 48.3M |
| GPS Rampášek et al. (2022) | 0.0858 | 19.4M |
| GEM-2 Liu et al. (2022) | **0.0806** | 32.0M |
| Graph Diffuser(ours) | 0.0867 | 20.3M |

Table 5: Evaluation on the recently suggested Peptides-func and Peptides-struct Dwivedi et al. (2022b).

| Model | # Params. | Peptides-func | Peptides-struct |
|---|---|---|---|
| | | AP $\uparrow$ | MAE $\downarrow$ |
| GCN | 508k | 0.5930±0.0023 | 0.3496±0.0013 |
| GINE | 476k | 0.5498±0.0079 | 0.3547±0.0045 |
| GatedGCN | 509k | 0.5864±0.0077 | 0.3420±0.0013 |
| GatedGCN+RWSE | 506k | 0.6069±0.0035 | 0.3357±0.0006 |
| Transformer+LapPE | 488k | 0.6326±0.0126 | 0.2529±0.0016 |
| SAN+LapPE | 493k | 0.6384±0.0121 | 0.2683±0.0043 |
| SAN+RWSE | 500k | 0.6439±0.0075 | 0.2545±0.0012 |
| Graph Diffuser | 509k | **0.6651±0.001** | **0.2461±0.001** |

Table 6: Comparison with the baselines in the PCQM-Contact dataset.

| Model | # Params. | Hits@1 $\uparrow$ | Hits@3 $\uparrow$ | Hits@10 $\uparrow$ | MRR $\uparrow$ |
|---|---|---|---|---|---|
| GINE | 517k | 0.1337±0.0013 | 0.3642±0.0043 | 0.8147±0.0062 | 0.3180±0.0027 |
| GCN | 504k | 0.1321±0.0007 | 0.3791±0.0004 | 0.8256±0.0006 | 0.3234±0.0006 |
| GatedGCN | 527k | 0.1279±0.0018 | 0.3783±0.0004 | 0.8433±0.0011 | 0.3218±0.0011 |
| GatedGCN+RWSE | 524k | 0.1288±0.0013 | 0.3808±0.0006 | 0.8517±0.0005 | 0.3242±0.0008 |
| Transformer+LapPE | 502k | 0.1221±0.0011 | 0.3679±0.0033 | 0.8517±0.0039 | 0.3174±0.0020 |
| SAN+LapPE | 499k | 0.1355±0.0017 | 0.4004±0.0021 | 0.8478±0.0044 | 0.3350±0.0003 |
| SAN+RWSE | 509k | 0.1312±0.0016 | 0.4030±0.0008 | 0.8550±0.0024 | 0.3341±0.0006 |
| Graph Diffuser(ours) | 521k | **0.1369± 0.0012** | **0.4053 ± 0.0011** | **0.8592± 0.0007** | **0.3388± 0.0011** |

## 6    CONCLUSION

In this work, we introduced a simple and effective architecture for machine learning on graphs, Graph Diffuser. Using a controlled example, we demonstrated its effectiveness in modeling long-range interactions while providing better interpretability. We then showed that this translates to real-world problems by evaluating GD on eight benchmarks from multiple domains. With minimal hyperparameter tuning, Graph Diffuser achieves a new state-of-the-art on most datasets and reaches very competitive results on the rest.

In the future, we plan to integrate Graph Diffuser with other promising Graph Transformer compositions, such as Transformers stacked on top of GNNs. Another area of interest is amplifying virtual edges, for example, by considering paths between nodes rather than just information propagation.

**Limitations** Our architecture integrates with the Transformer and consequently suffers from the quadratic memory complexity of the attention mechanism, which can restrict its applicability to larger graphs.

**REPRODUCIBILITY STATEMENT**    The authors support and advocate the principles of open science and reproducible research. We describe Graph Diffuser in detail in the text and figures and mention all relevant hyperparameters to reproduce our experiments in the appendix. Moreover, we will release our code as open-source with clear instructions on how to reproduce all of our results, as well as how to extend our model and apply it to new datasets.

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

## A   APPENDIX

### A.1   IMPLEMENTATION DETAILS FOR THE GRID HISTOGRAM COUNTING TASK

We generate 10k graphs and split them into 8k/1k/1k train/val/test sets. Each grid is of size $10 \times k$, where $k \in \{10, 11, 12, 13\}$. Node colors are drawn uniformly from a set of 20 colors. We search

over learning rates in $\{3e-4, 4e-4, 8e-4\}$, repeat each experiment with ten different random seeds and report the average and standard deviation of the best configuration. All models have a hidden dimension of 32. We use GatedGCN as GNN modules, and four attention heads in Transformer modules. Transformer models use six layers, $GNN \rightarrow Transformer$ uses 3 GNN layers and 3 Transformer layers. GNNs model has 12 layers to avoid underreaching. For Graph Diffuser, we found 3 layers sufficient. We use 0.2 dropout for GNN's modules and 0.5 dropout for attention. For the Diffuser model, we found no need for dropout.

## A.2 THE EFFECTS OF GD ON GRAPH TRANSFORMER VARIATIONS

We Conducted a simplified experiment to see the effects of GD when used as an out-of-the-box addition to different Transformer compositions, with no hyperparameter tuning. We evaluated 3 Transformer variations: *Transformer*, Transformer layers stacked on top of GNN layers ($GNN \rightarrow Transformer$), and Transformer interleaved with GNN in the same layer ($GNN \mid Transformer$). For each variation, we evaluate the unmodified architecture with no positional embedding or positional attention, the architecture with our positional encoding and attention added, but not learning the adjacency matrix(+ *Graph Diffuser*), and one we learn a new weighted adjacency matrix(+ *Weighted Adjacency*), as described in Section 4.1.3. The results are in Table 7.

Table 7: The effects of our architecture on different Graph Transformers. Adding positional encoding and attention improves all Transformer baselines. Learning a weighted adjacency improves GNN-Transformer combinations, but not the vanilla Transformer.

| Model | OGBG-PPA Accuracy ↑ | ZINC MAE ↓ |
|---|---|---|
| *Transformer* | 0.09483 | 0.7043 |
| *+ Graph Diffuser* | **0.7676** | **0.152** |
| *Weighted Adjacency* | 0.7750 | 0.1611 |
| *GNN → Transformer* | 0.3896 | 0.2068 |
| *+ Graph Diffuser* | 0.4954 | 0.1225 |
| *Weighted Adjacency* | **0.6372** | **0.1160** |
| *GNN │ Transformer* | 0.6793 | 0.1516 |
| *+ Graph Diffuser* | 0.7866 | 0.0880 |
| *Weighted Adjacency* | **0.7931** | **0.08671** |

## A.3 COMPUTATION ENVIRONMENT AND RESOURCES

Our implementation is based on the excellent open source Graph GPS implementation[3], which it self it based on Pyg and GraphGym Fey & Lenssen (2019); You et al. (2020). We use a machine with 24GB NVIDIA A10G GPU with 32GB or 64GB RAM for the larger datasets(ogbg-ppa, ogbg-code2,PCQM4Mv2), and a shared cluster equipped with NVIDIA TITAN Xp for the other datasets.

---

[3]https://github.com/rampasek/GraphGPS

Table 8: Graph Diffuser hyperparameters for the OGB benchmarks.

| Hyperparameter | ogbg-molpcba | ogbg-ppa | ogbg-code2 | ZINC |
|---|---|---|---|---|
| Transformer/GPS Layers | 5 | 3 | 4 | 10 |
| Edge-Wise FFN Layers | 2 | 2 | 2 | 2 |
| number of stacks(k) | 16 | 10 | 20 | 16 |
| MP-NN | GatedGCN | GatedGCN | GatedGCN | GINE |
| Hidden dim | 384 | 256 | 256 | 64 |
| Transformer FFN multiplier | x2 | x2 | x4 | x2 |
| Attention Heads | 4 | 8 | 4 | 4 |
| Dropout | 0.2 | 0.1 | 0.2 | 0. |
| Attention dropout | 0.5 | 0.5 | 0.5 | 0.5 |
| Graph pooling | mean | mean | mean | sum |
| Batch size | 512 | 32 | 32 | 32 |
| Learning Rate | 0.0005 | 0.0003 | 0.0001 | 0.001 |
| Epochs | 100 | 200 | 30 | 2000 |
| Warmup epochs | 5 | 10 | 2 | 50 |
| Weight decay | $1e-5$ | $1e-5$ | $1e-5$ | $1e-5$ |
| Parameters | $10646k$ | $3050k$ | $13912k$ | $443k$ |

Table 9: Hyperparameters for ZINC and the LRGB datasets.

| Hyperparameter | ZINC | PCQM-Contact | Peptides-func | Peptides-struct |
|---|---|---|---|---|
| Transformer/GPS Layers | 10 | 4 | 4 | 4 |
| Edge-Wise FFN Layers | 2 | 2 | 2 | 2 |
| number of stacks(k) | 16 | 16 | 16 | 16 |
| MP-NN | GINE | GatedGCN | - | - |
| Hidden dim | 64 | 92 | 112 | 112 |
| Transformer FFN multiplier | x2 | x2 | x2 | x2 |
| Attention Heads | 4 | 4 | 4 | 4 |
| Dropout | 0. | 0. | 0. | 0. |
| Attention dropout | 0.5 | 0.5 | 0.5 | 0.5 |
| Graph pooling | sum | sum | CLS | CLS |
| Batch size | 32 | 128 | 128 | 128 |
| Learning Rate | 0.001 | 0.0003 | 0.0003 | 0.0003 |
| Epochs | 2000 | 200 | 200 | 200 |
| Warmup epochs | 50 | 10 | 10 | 10 |
| Weight decay | $1e-5$ | 0 | 0 | 0 |
| Parameters | $443k$ | $503k$ | $509k$ | $509k$ |

