# OpenReview forum: "Diffusing Graph Attention"
_ICLR.cc/2023/Conference — Submitted to ICLR 2023_

### Official Review · Reviewer_6Q8a · 2022-10-23

**Confidence:** 3
**Correctness:** 2
**Technical Novelty And Significance:** 2
**Empirical Novelty And Significance:** 2
**Recommendation:** 3

**Clarity, Quality, Novelty And Reproducibility:**

The paper is mostly clear other than some parts mentioned above. To the best of my understanding, it seems the only novel part is generating a dense adjacency matrix from node and edge features before computing virtual node embedding and passing in graph transformer:
- using random walk matrices as relative embedding to modify the attention matrix (4.1.1, 4.2.1) is basically the same as the random walk kernel in GraphiT. Also Relative PE using random walks is also mentioned in GRAPHGPS where batch norm and MLP are added afterwards (corresponding to 4.1.2)
- using the diagonal of random walk matrices as positional embedding to add to the nodes feature (4.3) is same as RWPE (Graph Neural Networks with Learnable Structural and Positional Representations).


**Strength And Weaknesses:**

Strength
- The method is motivated by the failure of current architectures over a synthetic task. The proposed method brings significant improvement to it.
-  The method demonstrates empirical gains over some datasets, mostly on ones that requires capturing long-range interactions

Weaknesses
- Some important parts of the paper are confusing, mostly due to the ordering of the sections and the phrases:
    - It is not clear to me what "learn to combine information propagation over multiple different propagation steps in an
end-to-end manner" means. What are the "multiple different propagation steps"?
    - The use of the phrase "virtual edge" is somewhat confusing. Based on the later formulas, it seems these virtual edges are essentially generated features between all pair of nodes (based on 4.1), and replace the original edges/edge features. But the phrase itself and Figure 1 makes it seem like the virtual edges are added to the original set of edges.
- Some of the statements seem to be over-claimed:
    - "However, given the arbitrary structure of graphs, incorporating the input into the Transformer remains a challenging aspect in designing GTs, and so far, there has been no universal solution. We propose a simple architecture for incorporating structural data into the Transformer, Graph Diffuser (GD)..." It should be clarified that a lot of the current graph transformers designs are expressive enough incorporate arbitrary structure of graphs (e.g. Graphormer, GRAPHGPS, etc) and showed strong results. The authors should narrow down the claim to improving the modeling of long-range interaction.
    - "this work is the first Graph Transformer to [...] learn to combine information propagation over multiple different propagation steps in an end-to-end manner." It is not particularly clear to me what this statement means, but seems this is referring to feeding stacked random walk matrices through feed-forward neural networks in 4.1.2. I believe this has already been mentioned in GRAPHGPS and GraphiT (i.e. Relative PE using random walks).
- The first step of learning a dense adjacency matrix could be very expensive for large graphs.
- The empirical gain on ZINC and most of the OGB datasets seem quite small compared to the variance.


**Summary Of The Paper:**

This paper proposes a new Graph Transformer architecture, Graph Diffuser, to incorporate structural information in graphs, particularly the long-range interactions. Specifically, Graph Diffuser first generates a dense adjacency matrix from the node and edge features, then obtains the virtual edge features by feeding the concatenation of k random walk matrices (computed from the generated adjacency matrix) through a feed-forward neural network. The produced virtual edge features are then used both for positional encoding (by taking the self virtual edges) and modifying the attention matrix directly. Experiments over both a synthetic task and 8 benchmarks are performed to show the performance improvements.

**Summary Of The Review:**

Overall, the paper has limited novelty and contribution. The major novel contribution is improving long-distance interaction modeling by generating a dense adjacency matrix from node and edge features. But generating features between all pairs of nodes is inherently not scalable and even more expensive than the already expensive attention matrix.

---

### Official Review · Reviewer_6mkR · 2022-10-24

**Confidence:** 4
**Correctness:** 4
**Technical Novelty And Significance:** 2
**Empirical Novelty And Significance:** 3
**Recommendation:** 3

**Clarity, Quality, Novelty And Reproducibility:**

This paper is clearly written and provides extensive empirical analysis and results. However,  the technical contribution and novelty is limited.

**Strength And Weaknesses:**

Strength:
1. The proposed method performs favorably on benchmarks datasets.
2. The method is effective but is also quite simple in term of implementing it.
3. The counter example of the 2-d grid histogram counting problem for graph transformer is interesting as all state of the art model fail to solve it.

Weakness:
1. The technical contribution is incremental. Graph diffusion (both in discrete and the ODE form) has been studied and applied in the domain of graph neutral network. While adding graph diffusion to the positional attention is new, it is a trivial extension. [1][2][3]
2. While it is surprising that most graph transformer could not solve the grid based counting problem, this problem itself in the context of graph can be easily solved by diffusion matrix itself if node color is expressed as node feature with one-hot encoding.


[1] https://papers.nips.cc/paper/2019/hash/23c894276a2c5a16470e6a31f4618d73-Abstract.html
[2] https://aclanthology.org/2021.emnlp-main.642.pdf
[3] https://openreview.net/forum?id=EMxu-dzvJk

**Summary Of The Paper:**

The paper combines the graph diffusion with graph transformer models by updating the positional attention score with the edge representations attained from applying the MLP on top of the tensor for the diffusing operation. The proposed method performs favorably against other state of art models on real-world benchmark datasets. Particularly, the proposed method is able to solve the grid-histogram counting problem that other graph transformer fail to solve.

**Summary Of The Review:**

The paper proposed a new variant of graph transformer by encoding the graph diffusion tensor as edge features and feed into the positional attention score. The proposed model empirically performs well on benchmark dataset. However, the technical novelty is limited as it is quite incremental.

---

### Official Review · Reviewer_1vG8 · 2022-10-25

**Confidence:** 3
**Correctness:** 3
**Technical Novelty And Significance:** 2
**Empirical Novelty And Significance:** 2
**Recommendation:** 3

**Clarity, Quality, Novelty And Reproducibility:**

The clarity of the paper can be improved. The quality and novelty of the proposed method are limited. The authors haven't released their code to reproduce the results.

**Strength And Weaknesses:**

Pros:
The experiments can effectively show the performance of the proposed method.

Cons:
1. There are existing works [1, 2] about incorporating graph diffusion in attention-based GNN that are not covered in this work. Besides, in the Related Work section, the subsection "Diffusion" might be revised to "Diffusion in Graph" to avoid any ambiguity with "Diffusion Models".
2. The improvement on many benchmarks is limited compared to the standard deviations.
3. The models in Table 1 are not well described in the main context. For example, the settings of "GNN →Transformer" and "GNN | Transformer+RWPE" can be briefly mentioned in 5.1.1. Moreover, the necessity of designing "GNN →Transformer" and "GNN | Transformer+RWPE" for comparison is not clear.
4. About the figure drawing: Figure 1 is a little hard to understand since it contains terms like "Edge-Wise FFN/projection" and unclear attention maps that are only described in the latter sections. The Transformer architecture in Figure 3 is not well drawn. The Positional Attention module is even partly outside the Multi-Head Attention module.

[1] Wang, Guangtao, Rex Ying, Jing Huang, and Jure Leskovec. "Multi-hop attention graph neural network." arXiv preprint arXiv:2009.14332 (2020).

[2] Liu, Yonghao, Renchu Guan, Fausto Giunchiglia, Yanchun Liang, and Xiaoyue Feng. "Deep attention diffusion graph neural networks for text classification." In Proceedings of the 2021 Conference on Empirical Methods in Natural Language Processing, pp. 8142-8152. 2021.

**Summary Of The Paper:**

In this work, the authors propose a new way to learn positional encoding in Graph Transformers by extracting relationships between distant nodes in the graph. To evaluate the proposed approach, the authors design a simple graph task, Grid Histogram Counting, as well as use several benchmark datasets.

**Summary Of The Review:**

Overall, the proposed method solves the limitation of modeling long-range interactions in the current Graph Transformers. However, the novelty of the proposed method is limited given several existing related works. The effectiveness of the proposed method is not so significant on several benchmarks. The paper can be improved with more clear descriptions and illustrations.

---

### Official Review · Reviewer_NF2T · 2022-10-26

**Confidence:** 4
**Correctness:** 2
**Technical Novelty And Significance:** 3
**Empirical Novelty And Significance:** 2
**Recommendation:** 3

**Clarity, Quality, Novelty And Reproducibility:**

The description of methods and experiments are clear. The paper is easy to follow. The graph diffuser and context attention show the novelty of the proposed work.

However, in the reproducibility statement section, the authors states that the code will be available, and they will provide instruction on how to adopt other dataset into their code. However, It would be beneficial for the reviewer to judge the reproducibility of their proposed methods from the original implementation if they could release the source code and instruction on how to adopt the code to other dataset earlier.

**Strength And Weaknesses:**

Strength:
1. The paper presents two novel design for a graph transformer, the graph diffuser and context attention.
2. The paper provides many experiments in different datasets to demonstrate the performance of their approach.
3. The detail description of the experimental setup is also useful in understanding the benchmarks.

Weakness:
1. The authors should add more literature review. By googling "graph transformer", there is a survey [1] on arXiv that mentioned various methods regarding graph transformer. But some of those methods are not reviewed by the authors. Also, the experiments should include more baselines.
2. When use weighted adjacency, the model does not leverage any information from original adjacency.
3. More ablation study are needed to reveal the importance of each component. For example, context attention only, context attention + weighted adj, context attention + diffuser + weighted adj, and so on.

[1] Min, E., Chen, R., Bian, Y., Xu, T., Zhao, K., Huang, W., ... & Rong, Y. (2022). Transformer for Graphs: An Overview from Architecture Perspective. arXiv preprint arXiv:2202.08455. (https://arxiv.org/abs/2202.08455)

**Summary Of The Paper:**

This paper demonstrates a method called graph transformer on better integrating graph structures. The paper conducts a brief description on how there techniques is composed of and propose various of analysis on the effectiveness of graph diffuser on established benchmark.

**Summary Of The Review:**

My major concern of the paper is the leak of literature reviews. The paper should involve more methods regarding to graph transformers and compare the performance with them to show the advantages of the proposed work.

---

### Author Response · Authors · 2022-11-06
**Thank you**

We thank the reviewers for their time and helpful comments.

---

### Decision · Program_Chairs · 2023-01-20

**Decision:**

Reject

**Justification For Why Not Higher Score:**

There is a general consensus between all reviewers that major work is needed before this work would pass the bar for ICLR. The authors do not appear to contest this decision.

**Justification For Why Not Lower Score:**

N/A

**Metareview: Summary, Strengths And Weaknesses:**

The present work proposes Graph Diffuser, a method for diffusing graph attentional mechanisms in a way that promotes long-range interactions. I fully agree with all of the reviewers that the method is interesting, and the identified toy problem highly curious. However, all reviewers pointed out significant limitations in the claims of the work's novelty, as well as raising concerns about the relative outperformance compared to baseline methods. Since the authors did not provide a rebuttal, I am assuming they also agree with this assessment. I can only recommend rejection in the current form, but I highly recommend the authors to continue improving their work going forward, it has the potential to be valuable.